# Modulating Brain Activity with Invasive Brain–Computer Interface: A Narrative Review

**DOI:** 10.3390/brainsci13010134

**Published:** 2023-01-12

**Authors:** Zhi-Ping Zhao, Chuang Nie, Cheng-Teng Jiang, Sheng-Hao Cao, Kai-Xi Tian, Shan Yu, Jian-Wen Gu

**Affiliations:** 1School of Biological Science and Medical Engineering, Beihang University, Beijing 100191, China; 2Strategic Support Force Medical Center, Beijing 100101, China; 3Savaid Medical School, University of Chinese Academy of Sciences, Beijing 100049, China; 4Brainnetome Center and National Laboratory of Pattern Recognition, Chinese Academy of Sciences, Beijing 100190, China; 5School of Artificial Intelligence, University of Chinese Academy of Sciences, Beijing 100049, China; 6School of Future Technology, University of Chinese Academy of Sciences, Beijing 100049, China

**Keywords:** invasive, BCI, decode, encode, modulate, ICMS, DBS

## Abstract

Brain-computer interface (BCI) can be used as a real-time bidirectional information gateway between the brain and machines. In particular, rapid progress in invasive BCI, propelled by recent developments in electrode materials, miniature and power-efficient electronics, and neural signal decoding technologies has attracted wide attention. In this review, we first introduce the concepts of neuronal signal decoding and encoding that are fundamental for information exchanges in BCI. Then, we review the history and recent advances in invasive BCI, particularly through studies using neural signals for controlling external devices on one hand, and modulating brain activity on the other hand. Specifically, regarding modulating brain activity, we focus on two types of techniques, applying electrical stimulation to cortical and deep brain tissues, respectively. Finally, we discuss the related ethical issues concerning the clinical application of this emerging technology.

## 1. Introduction

Brain–computer interface (BCI) or brain–machine interface (BMI) is a gateway for information communication between the brain and computers that does not depend on the regular input and output pathways composed of sensory organs, peripheral nerves, and muscles [1]. It aims at restoring or enhancing brain functions by establishing an effective channel of information transmission between external devices, e.g., computers, and the nervous system.

Depending on how to recode neural signals, BCI is divided into two categories: non-invasive BCI and invasive BCI [2]. Non-invasive BCIs collect information about brain activity without requiring brain surgery by utilizing methods such as electroencephalography (EEG), magnetoencephalography (MEG), functional magnetic resonance imaging (fMRI), and functional near-infrared spectroscopy (fNIRS). The most popular non-invasive BCI technique is EEG, which records electrical brain activity signals using scalp electrodes. In contrast, invasive BCI records activity via surgically implanted electrodes close to the target neurons in the cortex or/and deep brain structures using, for example, microelectrode array (MEA), electrocorticography (ECoG) electrodes, stereo-electroencephalography (sEEG) electrodes, and deep brain stimulation (DBS) electrodes. MEA, which is typically placed within the gray matter of the cerebral cortex, is capable of detecting neuronal action potentials (also known as spikes) generated from a single or multiple neurons. ECoG electrodes (placed either below the dura matter, i.e., subdural, or above it, i.e., epidural) and sEEG electrodes record local field potentials (LFP), while DBS electrodes deliver electrical currents to activate or inactivate neuronal populations close by [3].

Invasive BCI, in comparison to non-invasive BCI, has essential advantages: (1) it can obtain neural signals with a much higher spatial and temporal resolution [4], e.g., to record activities from individual neurons or to modulate the activities of a small population of neurons; (2) it has a higher signal-to-noise ratio (SNR) and is more robust against electrical noise interferences or movement artifacts; and (3) its electrodes can be placed very close to or directly in the target cortical areas or subcortical structures, which is of paramount importance for developing BCI capable of decoding particular information and modulating specific brain functions. Of course, the disadvantages or limitations of the current invasive BCI technology are equally obvious. First, the implantation of electrodes directly into neural tissues requires a surgical operation that is invasive by itself and increases the risk of complications. Second, the system, once being implanted, is difficult to fix hardware problems or to update any component, therefore requiring much higher reliability and reducing a certain degree of flexibility. Finally, due to the complexity of the surgical procedure itself and the necessary care afterwards, invasive BCI is expensive, which needs to be addressed in order to increase its accessibility. Overall, the advantages of invasive BCI are more fundamental and its limitations may well be overcome through technological innovations. Thus, invasive BCI has great development potential and is a very promising technology for both basic neuroscience research and clinical application, particularly for modulating brain activity.

Performing a straightforward search in the MEDLINE (PubMed) database reveals a rapid expansion of the literature on invasive BCI, particularly in the past two decades. With a specific search formula: (((((brain-computer interface [Title/Abstract]) OR (brain machine interface [Title/Abstract])) AND (intracranial [Title/Abstract])) OR (invasive [Title/Abstract])) AND (cortex [Title/Abstract])) OR (deep brain [Title/Abstract]), 20,263 outcomes are available for the years 1956 to 2002. The results are shown in Figure 1a, indicating an obvious upward trend of published papers related to cortical and deep brain BCI since the 1950s. In Figure 1b, we summarize a timeline of important milestones and representative developments in this field.

Based on the direction of information flow between the brain and machines, invasive BCI can be further categorized into two classes: controlling external devices using recorded brain signals, sometimes called “control by the brain”, and modulating neural activities based on external signals, or sometimes called “modulation of the brain”. To achieve accurate “control by the brain” or “modulation of the brain”, it is fundamental to decode and encode neural activity correctly for understanding the meaning of brain signals, and how to modulate them for specific purposes. For example, by decoding the signals generated from brain activity, one can control external devices, including a cursor [20,21,22], robotic arm [16], etc., which has a great value for neural rehabilitation or functional compensation for patients with movement deficits due to stroke, or injury to the brain or the spinal cord [23]. On the other hand, by properly encoding external information to modulate brain activity, one can develop effective therapeutic techniques to deal with brain diseases, such as Parkinson’s disease, depression, etc., and reconstruct sensory function for people with visual or hearing disabilities.

For modulating brain activity by invasive BCI, according to the implant location of the electrodes, the technologies can be classed as cortical stimulation (focus on the cortex) and deep brain stimulation (focus on the deep brain tissues), which have been called intracortical microstimulation (ICMS) [24] and deep brain stimulation (DBS) [25], respectively. To date, ICMS has been mainly used to perform sensory feedback with high spatiotemporal precision [26,27,28], while DBS has been applied primarily to treat neurologic and neuropsychiatric disorders [29,30,31].

This review is aimed at helping researchers interested in BCI, particularly clinicians, to understand how invasive BCI can be used for modulating brain activities. To this end, we first introduce the concept of neural decoding and encoding, then discuss the progress in their applications by using invasive BCI, especially in modulating brain activity based on both ICMS and DBS. Moreover, the promising directions of invasive BCI, especially the closed-loop, real-time bidirectional BCI [32,33], are analyzed. Finally, the ethical issues related to clinical applications of invasive BCI are discussed.

## 2. The Framework of Neural Decoding and Encoding

Encoding and decoding are essential to understand how the brain works and interacts with outside environments, which is fundamental for accurate “control by the brain” or “modulation of the brain” by BCI. After Edgar Adrian successfully recorded the spikes produced by sensory nerves and reckoned that the firing rate of neurons could be an expression of external information [34], the study of neural decoding and encoding has become an essential field in neuroscience (see [35,36] for recent examples). In a nutshell, neural decoding and encoding are similar to looking up information in a dictionary. Decoding corresponds to retrieving the meaning of a specific word, while encoding corresponds to finding a specific word according to its meaning.

Specifically, the encoding of neural information typically refers to the representation of external stimuli by the activities of neurons. Decoding of neural information is to infer the characteristics of either external stimuli information or controlling commands from the observed neural activities. Decoding and encoding are closely connected and can be quantitatively formulated by the Bayesian theory. Let $P\left(s\right)$ represent the occurring probability of a specific external stimulus $\mathrm{and}\;P\left(r\right)$ represent the probability of observing a specific neural activity pattern. Thus, the likelihood of detecting activity pattern *r* when the stimulus *s* presents can be denoted as $P\left(r|s\right)$, known as conditional probability. Based on the Bayesian theory, we have
(1)$$P\left(s\right)P\left(r|s\right)=P\left(r\right)P\left(s|r\right)$$

To solve the neural encoding problem is to calculate $\left(r|s\right)$, which can be computed as
(2)$$P\left(r|s\right)=\frac{P\left(r\right)P\left(s|r\right)}{P\left(s\right)}$$

To solve the neural decoding problem is to calculate $P\left(s|r\right)$, which can be computed as
(3)$$P\left(s|r\right)=\frac{P\left(s\right)P\left(r|s\right)}{P\left(r\right)}$$

The above-mentioned analysis can be directly extended to calculate the distribution of neural activities, namely:(4)$$P\left(r\right)=P\left(r|s_{1}\right)P\left(s_{1}\right)+P\left(r|s_{2}\right)P\left(s_{2}\right)+\cdots +P\left(r|s_{n}\right)P\left(s_{n}\right)=\overset{n}{\underset{n=1}{\sum }}P\left(r|s_{n}\right)P\left(s_{n}\right)$$
where *s*_n_ represents the *n*^th^ external stimulus, and the external stimuli, namely:(5)$$P\left(s\right)=P\left(s|r_{1}\right)P\left(r_{1}\right)+P\left(s|r_{2}\right)P\left(r_{2}\right)+\cdots +P\left(s|r_{i}\right)P\left(r_{i}\right)=\overset{i}{\underset{i=1}{\sum }}P\left(s|r_{i}\right)P\left(r_{i}\right)$$
where *r*_i_ represents the *i*^th^ neural activity pattern.

Based on the encoding–decoding framework described above, modulating neural activities for a specific purpose, for instance, hacking the brain to mistakenly perceive a specific stimulus *s* which does not actually exist, is to modulate the activity pattern to resemble *r*, which satisfies
(6)$$r=arg\;\underset{r}{max}\;P(s|r)$$

Similarly, to decode the meaning of activity pattern *r* is to find the stimulus *s* that satisfies
(7)$$s=arg\;\underset{r}{max}\;P(r|s)$$

## 3. Application of Decoding for Controlling External Devices

Various invasive BCIs have been developed to control a robotic arm by decoding the neural activity of the primary motor cortex (M1) [16]. It is a promising way to help disabled persons improve their quality of life by inserting electrodes close to the neurons of brain tissues where the spikes of neurons’ firing can be recorded with ~1 ms resolution. Different from turning on/off external devices [37], it is desirable that “free manipulation” of robotic arms via invasive BCIs can be achieved, for which appropriate decoding methods are the key. Many decoding methods are effective in predicting arm movements, such as population vector algorithms [38], optimal linear estimators [39], sliced inverse regression [40], the Bayesian decoder [41] and Kalman filter [42], which are used to predict the movement direction, speed, and trajectory. Besides decoding accuracy, another important factor needs to be considered: the ability to control robotic arms in real-time.

The real-time control was developed by decoding M1 activities in monkeys and using them to control a cursor movement without preliminary training [43,44]. Schwartz et al. optimized the decoding of the monkey’s M1 to achieve more precise manipulation of the robotic arms, which included movement in the 3D space and gripping force [15]. In humans, a tetraplegic patient controlled the cursor movement on a screen in real time with the assistance of invasive BCI [45], which was proved chronically safe [46]. In 2012, a patient with tetraplegia successfully drank coffee by using an invasive BCI to control the robotic arm, which was a breakthrough in clinical application [16]. Recently the invasive BCI has been used to help patients use a bimanual robotic limbs system [47]. Although a robotic arm controlled by invasive BCIs will be a replacement for rehabilitation of the lost motor function in the future, it is necessary to combine it with perceptual feedback for better control [48]. Invasive BCIs with perceptual feedback and bidirectional, closed-loop motor control will be progressed using ICMS and DBS technologies.

Recently, invasive BCIs have been developed to accomplish other functions along with controlling robotic arms (see Figure 2). Capogrosso et al. used an invasive BCI to alleviate the gait deficits of macaques with a spinal cord injury by decoding the motor states from M1 and transmitting them to an electrical stimulator in the spinal cord [17]. Gopala et al. used an invasive BCI to synthesize the correct artificial speech when the subjects imitated sentences silently by recurrent neural networks to decode the neural activity of the cortex into a representation of articulatory movement and transform it into speech acoustics (see Figure 2) [49]. Francis et al. decoded the intention of a patient who was trying to write letters from the neural activity of the motor cortex by the recurrent neural network and converted it into text in real time by implanting microelectrode arrays into the motor cortex. Using this invasive BCI, patients achieved a typing speed of 90 characters/min, with an online original accuracy of 94.1% and offline automatic correction accuracy of more than 99% [50], while the typing speed on typical smartphones is about 115 characters/min (see Figure 2). Dekleva et al. developed a new decoding approach for cursor point-and-click based on identifying hand grasp, which provides a high-performance cursor control [51] (see Figure 2). It is noteworthy that, in recent decoding studies, the methods based on artificial neural networks have attracted much attention [52,53], as they provide a promising way to combine deep learning with BCI to produce accurate estimation/prediction using spikes [54,55,56,57] or intracortical LFPs [58,59,60,61].

## 4. Modulating Cortical Activity by ICMS

Ever since Wilder Penfield mapped the “cortical homunculus” in the motor cortex using electrical stimulation [62], many scholars have used ICMS technology to investigate more precise motor cortex and sensory cortex maps [63,64,65] and have pointed out a clear direction to regulate visual–motor function. As a method for injecting information into the cortex, Salzman et al. used ICMS to stimulate selective neurons in the middle temporal visual area (MT or V5) to change the visual motor information processing in macaques, making their eyes more oriented towards the motion encoded by the stimulated neurons [66]. Fujii et al. used ICMS to stimulate the frontal eye field (FEF) of macaques to interfere with the gaze task. They found that stimulating unilateral FEF and bilateral FEF induced a reverse saccade and a two-step saccade, respectively [67]. Overall, ICMS has been proven as an effective method to understand the functional role of cortical activity and establish a basis for manipulating sensory perception through modulating cortical activity by invasive BCIs.

### 4.1. ICMS for Restoration of Tactile Feedback

The fundamental research on the primary somatosensory cortex (S1) by ICMS is vital for invasive BCIs with perceptual feedback. Romo et al. trained two macaques to distinguish between two mechanical flutter stimuli applied sequentially to the fingertips. Microelectrodes were inserted into clusters of quickly adapting (QA) neurons of the S1, and the first or both stimuli were then substituted with trains of current pulses during the discrimination task. It was found that microstimulation can be used to elicit a memorizable and discriminable analog range of percepts, and the activation of the QA circuit of S1 was sufficient to initiate all the subsequent neural processes associated with flutter discrimination [68]. Recently, Flesher et al. used ICMS to evoke the hand sensation of patients with long-term spinal cord injuries. By applying ICMS in the hand area of the somatosensory cortex, they successfully evoked the pressure sensitivity of all parts of the fingers and maintained stability for a long time (see Figure 3a). In addition, adjusting the stimulus amplitude classified the perceptual intensity of the stimuli, indicating that ICMS could be used to convey information about the contact position and pressure required for hand movement [69]. In addition, chronic ICMS may be a viable means of transmitting sensory feedback into neural prostheses without inducing significant damage [70].

Using ICMS, an invasive BCI could be a brain–machine–brain interface (BMBI) [71], which controls the exploratory arrival movement of the actuator and facilitates artificial tactile feedback. In 2009, Doherty et al. trained macaques to move the cursor on the screen by hand, then to control the virtual robotic arms through the neural activity in M1 and dorsal premotor cortex (PMd) and established a correlation between touching virtual objects and ICMS in S1. The macaques successfully underwent BMBI to find and distinguish three visual objects and successfully identified each object through the virtual robotic arm [48]. Richard et al. also demonstrated that BMBI with ICMS in S1 can provide tactile sensation [72]. Recently, Flesher et al. supplemented vision with tactile perception evoked using a bidirectional BCI that records neural activity from the motor cortex and generates tactile sensations through intracortical microstimulation of the somatosensory cortex, enabling subjects with tetraplegia to substantially improve performance with a robotic limb [73]. For amputees, a neural–machine-interface prosthesis with sensory feedback directly on a missing limb in real-time is widely used [74,75,76,77,78].

### 4.2. ICMS for Restoration of Visual Sense

Along with restoring tactile feedback by ICMS in S1, some scholars attempted to modulate the visual cortical activity to restore the visual sense. In 1968, Brindley and Lewin implanted an electrode array in the occipital lobe of the right hemisphere of a blind patient, and she could then feel light in the left half of the field of vision after ICMS [79]. Dobelle et al. found that ICMS could produce a sense of visualizing light spots (phosphenes), which appeared immediately at the beginning of stimulation and disappeared immediately after cessation of stimulation [80]. Schmidt et al. found that the visual sensation produced by ICMS was more evident in the report of a blind patient, and even color could be produced [81]. Edward et al. used ICMS in the primary visual cortex (V1) of macaques, revealing that the size of the phosphene depended on the current transmitted to V1 and the retinal cortical magnification factor [82]. These findings are valuable in the direction of the application of ICMS in the visual cortex for regaining sight and developing visual prostheses [83].

Recently, Chen et al. successfully realized the modulation of the activity of the visual cortex through invasive BCI. They implanted Utah electrodes (a total of 1024 channels) in the V1 and V4 regions of the visual cortex of monkeys and used ICMS to trigger optical illusions by injecting currents through hundreds of electrodes. Meanwhile, the positions of these electrodes were mapped with the receptive fields of stimulated neurons. Finally, they encoded the patterns composed of several optical illusions and simultaneously stimulated multiple visual cortical neurons through ICMS. Monkeys could immediately recognize them as simple shapes, letters, etc. [84]. Recently, a 96-channel microelectrode array was implanted into the visual cortex of a 57-year-old patient with complete blindness for 6 months. The research team developed an invasive BCI with ICMS in the occipital cortex and successfully helped the patient to identify some letters and object boundaries (see Figure 3b) [85]. These results confirmed the potential of invasive BCI for restoring functional vision.

## 5. Modulating Brain Activity by DBS

Conventionally, DBS is composed of an electrode and a pulse generator whose stimulation parameters, including amplitude, frequency, pulse duration, and pulse width, are controlled by a computer (see Figure 4). However, unlike ICMS, whose effective stimulation current ranges from 40 to 180 μA to target a relatively small number of neurons, the effective stimulation current of DBS ranges from 1 to 40 mA [86], aiming at manipulating a large group of neurons. Scholars have mainly used DBS to treat neurologic and neuropsychiatric conditions, including essential tremor (ET), dystonia, Parkinson’s disease (PD), and refractory obsessive-compulsive disorder (OCD) [87], which have been approved by the Food and Drug Administration (FDA).

### 5.1. DBS for Neurological Disorders

Brice et al. found that the DBS for contralateral midbrain and basal ganglia could inhibit intentional tremor in three patients, and then they implanted permanent electrodes in two patients. During the half-year follow-up, the therapeutic effect of the DBS remained without changing the electrical threshold [11]. Benabid et al. relieved tremors in 26 patients with PD and 6 patients with ET through high-frequency (130 Hz) stimulation of the ventral intermediate nucleus (VIM) [88]. They found that VIM-DBS could effectively inhibit tremors and replace thalamic surgeries in the treatment of PD and ET [12]. They also noted that DBS for the subthalamic nucleus (STN) could reduce the dosage of administrated medicines by 30–50%, and DBS for the globus pallidus internus (GPi) could inhibit abnormal involuntary movements [13]. Recent studies have also shown that the GPi-DBS could effectively treat dystonia [89], Huntington’s disease [90], and Tourette syndrome [91]. Although the therapeutic effect of DBS on tremors is noticeable, the underlying mechanism has remained elusive. Benazzouz et al. found that high-frequency stimulation of the STN reduced the excitatory glutamate output, resulting in the inactivation of the reticular neurons of black substance, which may be the mechanism of inhibiting tremors by DBS [92]. Boraud et al. found that high-frequency stimulation of GPi could significantly reduce the high discharge rate of GPi, thereby alleviating the tetany and motor symptoms of PD [14]. Some researchers also pointed out that high-frequency stimulation of GPi may interrupt the abnormal pattern of thalamic discharge [93]. Common targets of DBS for tremors include STN, GPi, and VIM (see Figure 4) [94,95]. Besides, DBS has been recently applied to treat refractory epilepsy [96,97].

### 5.2. DBS for Neuropsychiatric Disorders

In recent years, DBS has also been used for treatment-resistant depression (TRD), addiction, anorexia, OCD, schizophrenia, and other mental diseases, which has shown good efficacy when targeting the nucleus accumbens (NAc) [98,99,100,101]. Recent research has shown that DBS in the internal capsule [102] and caudate nucleus [103] effectively treat OCD (see Figure 4). In addition, DBS in the amygdala can effectively alleviate post-traumatic stress disorder [104,105,106]. Castillo et al. found that DBS in the globus pallidus externus can effectively treat insomnia in patients with PD [107]. Recently, Scangos et al. have effectively treated severe depression through a closed-loop invasive BCI. In this study, researchers first identified specific biomarkers of depression and potential treatment sites by observing deep brain signals. Then, a closed-loop BCI system was developed by using the biomarkers detected in the brain signals to modulate the stimulators chronically implanted in the deep brain area, achieving controlled regulation of the biomarkers. The depressive symptoms of the patient were relieved during the one-year period of the treatment [19]. It is noteworthy that developing closed-loop invasive BCI will be one of the leading research directions for the clinical treatment of some neuropsychiatric disorders [108].

## 6. Technological Challenges and Future Directions

Although this review focuses on invasive BCI techniques, it is noteworthy that cortical and deep brain regions can also be stimulated at the macroscopic level using non-invasive methods, such as transcranial magnetic stimulation (TMS), transcranial direct current stimulation (tDCS), and transcranial focused ultrasound (tFUS), in order to treat neurodegenerative diseases and neuropsychiatric disorders [109,110,111,112]. A new method of stimulating cortical regions at the microscopic level is optogenetics. It works by utilizing light to activate neurons that have been either genetically manipulated or infected by specific viruses to express light-sensitive ion channels, which is currently only used in animal research [113,114].

ICMS and DBS, particularly DBS technology, have been used with success to treat certain neuropsychiatric and neurological disorders. However, there are still two main significant technical obstacles to be overcome for the development of invasive BCIs: (1) to minimize tissue damage due to electrode implantation, (2) to prolong the periods in which electrodes can effectively work in vivo. Currently, invasive BCI, especially ICMS-based technology, still needs complicated craniotomy to temporally remove a relatively large piece of skull, which adds more risks to both the surgery itself and later recovery. Novel implantation techniques that can minimize the wound and simplify the procedure will be very helpful. In addition, widely-used silicon-based electrodes irritate the surrounding tissues after implantation. In the longer term, this can cause cellular responses that change the local environments, resulting in (1) encapsulation of the electrodes by glia cells and deterioration of the recorded signals, and (2) neural cell death close to the implantation site [3]. Materials that are more biocompatible will be necessary for the development of electrodes in the future, e.g., the flexible electrode which reduces damage to brain tissue and foreign body reactions [115]. Neuralink recently demonstrated flexible electrodes named “treads”, which are inserted in the cortex through specially designed surgical robots [18]. Another interesting electrode introduced recently is called a “stentrode”, which is delivered endovascularly to be close to the target area to record signals [21]. Similar techniques may also be used for DBS [116].

## 7. Ethical Issues of Invasive BCI

Clinical research on invasive BCI is currently being used to treat diseases that cannot be treated with drugs, such as paralysis, muscular dystrophy, and others, by allowing patients to control assistive devices, regain important daily motor functions or lost sensations. It can also be used as a complementary treatment to drug-based therapies for neurological and neuropsychiatric disorders. Although invasive BCI has enormous potential in clinical applications, its surgical complications, such as bleeding, immunologic rejection, infection, and mechanical failure [117], need to be treated with more caution. In particular, the precise localization that is important for electrode implantation and the comprehensive understanding of neural circuits, which is important to modulate brain activity through invasive BCI, are still potential problems that may induce side effects over time. In addition, other factors, such as the psychological acceptance of the patient for invasive BCI, privacy, etc., need to be weighed. The following notes should be considered in the clinical application of invasive BCI: 1. The specific medical benefits, such as diagnosis, cure, palliative treatment, or prevention. 2. Possible damage and side effects. 3. Assistance to the patients to better understand invasive BCI and improve their psychological acceptance. 4. Assistance to the patients to more accurately understand the prognosis after BCI treatment. 5. Providing solutions for the failure of invasive BCI.

Currently, invasive BCI is in the early stages, especially in modulating brain activity. In the future, with the development of decoding and encoding research, electrodes, implantation methods, and other technologies, as well as the mature encoding and decoding technologies of brain activity, invasive BCI can effectively read and even “write” information to the brain. It will face the risks of privacy disclosure and malicious interference [118]. Therefore, designing a series of hardware and software is essential to protect the safety and security of invasive BCI. As suggested by Zeng et al., it should be people-oriented and protect user privacy. Comprehensive and strict standards should be established for identity authentication, information encryption, and system protection. Besides, it is crucial to prohibit abuse and misuse, and develop intrusion detection systems as well as the corresponding protection mechanisms [119]. The formulation of ethical principles is the key to the responsible development of invasive BCI in clinical applications.

## Figures and Tables

**Figure 1 brainsci-13-00134-f001:**
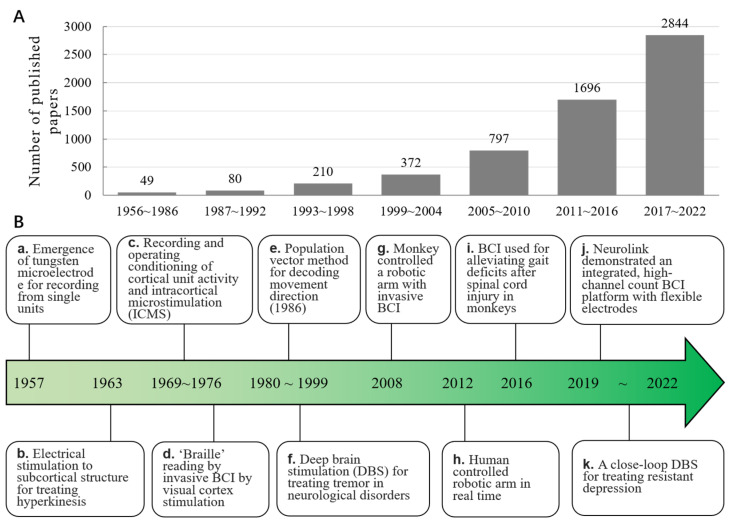
The progression of invasive BCI. (**A**). The number of published papers over time. A rapid growth in the number of papers in recent (20) years demonstrates the rapidly increasing interest in invasive BCI technology. (**B**). The historical timeline for major breakthroughs and representative developments in invasive BCI. The references cited in **a**–**k** correspond to the following: **a** [5], **b** [6], **c** [7,8], **d** [9], **e** [10], **f** [11,12,13,14], **g** [15], **h** [16], **i** [17], **j** [18], and **k** [19].

**Figure 2 brainsci-13-00134-f002:**
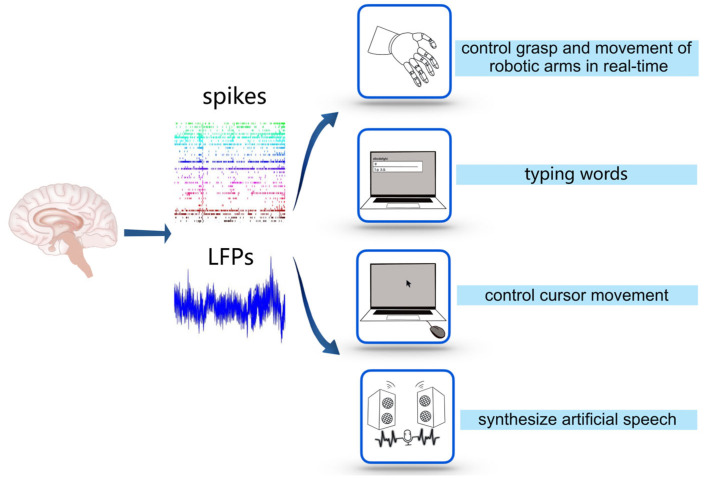
Applications of decoding. The application of decoding is a process from recording electrical signals of brain activity, such as spikes and LFPs, to using them to control external devices.

**Figure 3 brainsci-13-00134-f003:**
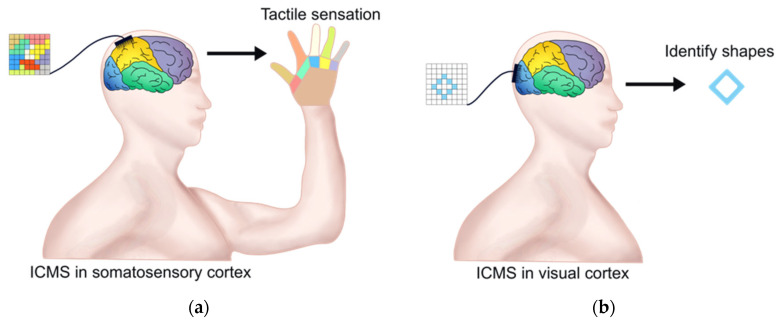
ICMS in restoring tactile sensation and vision. (**a**) ICMS with the microelectrode array in the primary somatosensory cortex can evoke tactile sensation of hand. The left colorful square represents the microelectrode array and the color is the ICMS area relative to tactile sensation of the hand area with the same color; (**b**) ICMS with microelectrode array in the visual cortex can help patients to identify some letters and object boundaries. The left square represents the microelectrode array and the blue rhombus within it represents the ICMS area, which relates to the optical illusion.

**Figure 4 brainsci-13-00134-f004:**
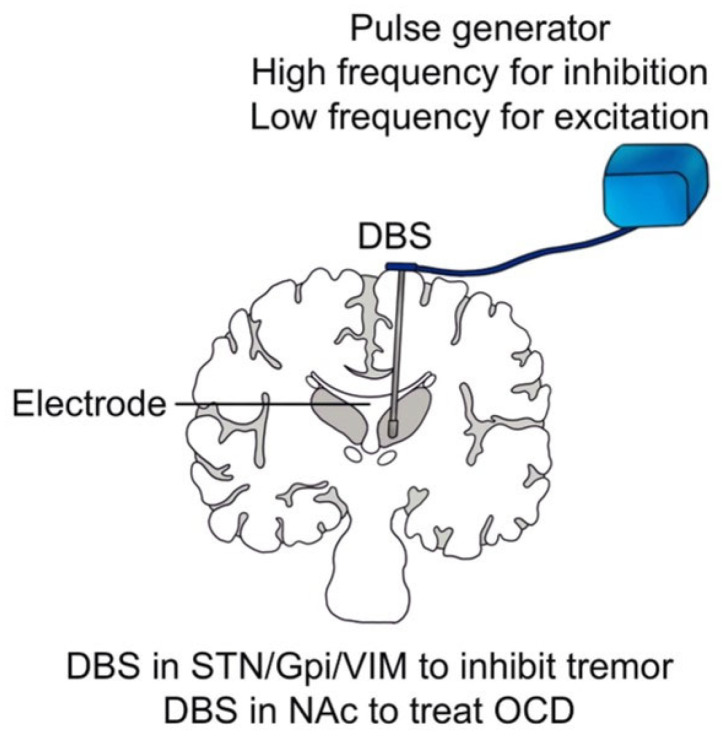
Framework and clinical application of DBS. Framework and clinical application of DBS. The blue bag represents the pulse generator and the stick in brain tissue represents electrode of DBS.

## Data Availability

No data were used to support this study.

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
