# Peer review of "Modulating Brain Activity with Invasive Brain–Computer Interface: A Narrative Review"

_brainsci, 2023, doi:10.3390/brainsci13010134_

Round 1

Reviewer 1 Report

Dear Editor and Authors

Congratulations on your research. The objective of the research responds to the need to employ brain-computer interfaces that respond to human needs. In this sense, a review is conducted. Although, it is not a systematic review following prism. Even so, I find the contributions very interesting and it is very well structured. I only miss two aspects. That in the introduction meta-analysis studies and systematic reviews are named, this would give it more solidity. And secondly, a section on methodology explaining, if it is a review, which Boolean actions were used and in which databases, how the inclusion and exclusion criteria were applied.  

Author Response

Thanks for your helpful suggestions. This review was a narrative review rather than a systematic review based on a comprehensive reference search and meta-analysis. Because the primary goal of this article is to help researchers interested in BCI, particularly clinicians, gain a comprehensive and systematic understanding of how invasive brain-computer interfaces can be used for brain activity modulation. To make this point clear, we have changed the title to “Modulating Brain Activity with Invasive Brain-Computer Interface: A Narrative Review” and revised the texts in Lines 86-87.

Based on your suggestions, we have also made added a comprehensive literature search to show that invasive brain-computer interfaces are receiving more attention in recent years. We provided search strategies and the results are shown in the added Figure 1. a and Line 56-62, which indicate a clear upward trend in number of published studies.

Reviewer 2 Report

Literature survey was not sufficient.

Please discuss some more methods available in current research.

make the comparision between  Invasive Brain-Computer Interface and  non Invasive Brain-Computer Interface .

A review analyzed for this study was not sufficient.

Author Response

Response 1:  We appreciate your suggestions. We add a ‘Technological challenges and future direction’ with two paragraphs in Lines 287~309. In this part, we add more methods such as TMS, tDCS, tFUS, and optogenetic simulation currently used or under investigation, in Lines 287~294, as well as the most up-to-date electrodes of invasive BCI technology such as ‘Treads’ and ‘Stentrodes’ in Lines 304~309.

Response 2: We added the comparison between non-invasive and invasive BCI, see Lines 37~49 and Lines 52~55.

Response 3: As we explained to reviewers 1 and 6, this review was a narrative review rather than a systematic review based on a comprehensive reference search and meta-analysis. Because the primary goal of this article is to help researchers interested in BCI, particularly clinicians, gain a comprehensive and systematic understanding of how invasive brain-computer interfaces can be used for brain activity modulation. To clarify this, we have changed the title to “Modulating Brain Activity with Invasive Brain-Computer Interface: A Narrative Review” and revised the texts in Lines 85-86. Based on reviewers’ suggestions, we have also made added a comprehensive literature search to show that invasive brain-computer interfaces are receiving more attention in recent years. We provided search strategies and the results are shown in the added Figure 1. a and Lines 56-62, which indicate a clear upward trend in a number of published studies.

Reviewer 3 Report

The authors provide a concise overview of the burgeoning field of brain computer interface. The text will be useful as a quick read summary for beginners as well as a rapid review for more experienced researchers in this field. I would recommend the authors to add an image covering the historical timeline for major breakthroughs and incremental improvements in this field. Overall it is a nice review suitable for publication in this journal. 

Author Response

Reply:

We are very grateful to your positive comments and helpful suggestions. We added the historical timeline for major breakthroughs and incremental improvements in this field in figure 1.b.

Reviewer 4 Report

The topic revised in this work is interesting and evolves rapidly alongside technology. The document is readable and describes many references. However, I found some concerns about the proposed work:

1)      I could not find specific statements about the advantages of using different approaches to Intracortical Microstimulation (ICMS) and Deep Brain Stimulation (DBS), in the sense of how previous works improve the accuracy results or the efficiency during the coding and decoding process (chronologically)? That is, most of the sentences are limited to mentioning that accurate estimation or prediction is achieved (lines 153 and 154). However, eight references are cited, and any description is proposed to differentiate the field’s advantages or improvements.   

2)      Similarly, in lines 198, 261, and 263, any further comparison is provided among the cited works. I recommend including a Table of comparison to highlight important differences and contributions of the revised works.

3)      In section 5, the authors provide a list of considerations in the clinical application, but first, it should be described what is currently established in the clinical application of invasive BCI. What should be avoided when we perform tests with patients?

4)      A conclusion should state how previous works improve the accuracy results or the efficiency during the coding and decoding process. Moreover, it is necessary to highlight the open issue of short-term and long-term for both contexts, ICMS and DBS. This is not clearly described.

5) Finally, I have one question: why this review is essential for authors? What context will the authors explore in future works about BCI invasive sensors? That should be interesting to understand why the 3.1 and 3.2 subsections are reviewed for ICMS and 4.1 and 4.2 for DBS.

Author Response

Reply:

  1. Response 1: We sincerely appreciate your suggestions. We added the comparison between non-invasive and invasive BCI to explain the advantage of invasive BCI in modulating brain activity which mainly from the high signal/noise ratio and finer spatiotemporal resolution. Please see Lines 37~49 and Lines 52~55 where the text was added in. We are not aiming to compare the differences among the cited 8 papers. Instead, they were included to give interested readers an index to look for further information.
  2. Response 2: Similar to the above point, these references are intended to illustrate the current trend of combining deep learning with either spike or LFP recordings, by giving relevant examples.
  3. Response 3: In the section “Modulating Brain Activity by ICMS and DBS”, we showed the currently established guideline in the clinical application of invasive BCI. And in the section “Ethical issue of invasive BCI”, we explained the problems of it in clinical application, e.g., Lines 208~209, Line 230, Lines 242~245, and Lines 267~270. In addition, we add a description of the clinical application of invasive BCI in the “6. Ethical Issues of Invasive BCI” section in Lines 311~315.
  4. Response 4: We added the comparison between non-invasive and invasive BCI and explained the advantage of invasive BCI in modulating brain activity, see Lines 37~49 and Lines 52~55. In addition, we add the section ‘technological challenges and future direction’, where we describe the open issues of electrodes chronically implanted of invasive BCI, see Lines 294~303.
  5. Response 5: The primary goal of this article is to help researchers interested in BCI, particularly clinicians, gain a comprehensive and systematic understanding of how invasive brain-computer interfaces can be used for brain activity modulation. As ICMS and DBS are two different approaches that can be used to address different clinical conditions, e.g., ICMS for sensory reconstruction and DBS for treating neurological and neuropsychiatric disorders, we introduce them separately in the paper. In the future, we will explore the clinical applications of invasive BCI.

Reviewer 5 Report

I have reviewed this manuscript the overall contents of this manuscript is well organized to give a clear overview of this work. I have suggested some comments about this work are as the following:

Comments to the Authors:

1.     Authors should add one table related to different type of BCI methods like, MI, P300, and SSVEP with model performance accuracy.

2.      My suggestion is that the authors should write discussion section clearly in more details like how and why BCI is important for clinical research.

3.     Author should write some limitation of BCI system, like to control BCI system, how to stop ongoing commend.

4.     Authors should add some latest reference about the BCI studies, like  

Author Response

Reply:

  1. Response 1:We really appreciate your precious advice. We added the comparison between non-invasive and invasive BCI to explain the advantage of invasive BCI in modulating brain activity which mainly from the electrode signal quality and finer spatiotemporal resolution. See Line 37~49 and Line 52~55. However, as the current paper is focused on invasive BCI, we did not elaborate on non-invasive approaches.
  2. Response 2:In the revision, we explained the value of invasive BCI for healing diseases in clinical applications, such as using ICMS for restoration of tactile and visual sensation, for which no drug can help, and treating neurological disorders and neuropsychiatric disorders, for which currently used drugs have long term side effects. In addition, we added this description in “6. Ethical Issues of Invasive BCI” in lines 311~315.
  3. Response 3:We add the section ‘Technological challenges and future direction’, where we describe some limitations of invasive BCI system and its future development (see Line 293~309).
  4. Response 4:In the new section “Technological challenges and future direction” part, we add some more methods such as TMS, tDCS, tFUS and optogenetic simulation, in Line 285~292, as well as the most up-to-date electrodes of invasive BCI technology such as ‘Treads’ and ‘Stentrodes’ in Line 306~309 with latest references.

Reviewer 6 Report

The paper reports a literature review regarding the invasive brain computer interface (BCI). The paper is of great interest and well written, however, in my opinion, some concerns need to be addressed before publication:

  • The main concern is the search for literature. Please, provide some information regarding the keywords employed, the databases searched, specify whether the search was conducted by different operators, and specify the criteria for selection of the papers reported in the manuscript. Although this is a narrative review, the Authors could refer to some guidelines for reviews, such as those reported below:
    • Rethlefsen, M. L., Kirtley, S., Waffenschmidt, S., Ayala, A. P., Moher, D., Page, M. J., & Koffel, J. B. (2021). PRISMA-S: an extension to the PRISMA statement for reporting literature searches in systematic reviews. Systematic reviews10(1), 1-19.
    • Kitchenham, B. (2004). Procedures for performing systematic reviews. Keele, UK, Keele University33(2004), 1-26.
  • It would be interesting if the Authors could provide some information regarding the kinds of electrodes and materials employed for invasive BCI, highlighting the issues related to the implants.
  • Although some future directions are already reported in the manuscript, in my opinion, a conclusion section should be added in order to specify the current achievements and limitations of this technique and the issues related to future developments.

Author Response

Reply:

  1. Response 1: We genuinely appreciate your constructive suggestions. As we explained to reviewers 1 and 2, this review was a narrative review rather than a traditional systematic review, neither selection nor meta-analysis was utilized. Because the primary goal of this article is to help researchers interested in BCI, particularly clinicians, gain a comprehensive and systematic understanding of how invasive brain-computer interfaces can be used for brain activity modulation. To make this point clear, we have changed the title to “Modulating Brain Activity with Invasive Brain-Computer Interface: A Narrative Review” and revised the texts in Lines 87~88. And we have also made added a comprehensive literature search to show that invasive brain-computer interfaces are receiving more attention in recent years. We provided search strategies and the results are shown in the added Figure 1. a and Line 56~61, which indicate a clear upward trend in a number of published studies.
  2. Response 2: We add the section ‘Technological challenges and future direction’, where we describe some limitations of invasive BCI system, as well as its future evolution, see Line 284~309. In these parts, the kinds of electrodes and issues related to the implants were also discussed in lines 294~306.
  3. Response 3: According to your suggestion, we added the section ‘Technological challenges and future direction’ in Lines 284~309, where we specify this technique's current achievements and limitations and the issues related to future developments.

Round 2

Reviewer 1 Report

Dear Authors and Editor, 

Perfect! Go ahead

Author Response

Thank you very much.

Reviewer 2 Report

Paper work was improved by the authors.

Author Response

Thanks for your suggestion again.

Reviewer 6 Report

I thank the Authors for addressing my comments. In  my opinion, the paper is improved and it is suitable for publication in the present form.

Author Response

We really appreciate your suggestions earlier. Thank you again for your comments.